# Validity of the Food Frequency Questionnaire—Estimated Intakes of Sodium, Potassium, and Sodium-to-Potassium Ratio for Screening at a Point of Absolute Intake among Middle-Aged and Older Japanese Adults

**DOI:** 10.3390/nu14132594

**Published:** 2022-06-23

**Authors:** Tomoka Matsuno, Ribeka Takachi, Junko Ishihara, Yuri Ishii, Kumiko Kito, Sachiko Maruya, Kazutoshi Nakamura, Junta Tanaka, Kazumasa Yamagishi, Taiki Yamaji, Hiroyasu Iso, Motoki Iwasaki, Shoichiro Tsugane, Norie Sawada

**Affiliations:** 1Department of Food Science and Nutrition, Graduate School of Humanities and Sciences, Nara Women’s University, Kitauoyahigashimachi Nara-shi, Nara 630-8506, Japan; renokkuma.22.sora@gmail.com (T.M.); s-maruya@cc.nara-wu.ac.jp (S.M.); 2Graduate School of Environmental Health, Azabu University, 1-17-71 Fuchinobe, Chuo-ku, Sagamihara-City 252-5201, Kanagawa, Japan; j-ishihara@azabu-u.ac.jp; 3Division of Cohort Research, National Cancer Center Institute for Cancer Control, 5-1-1 Tsukiji, Chuo-ku, Tokyo 104-0045, Japan; yurishii@ncc.go.jp (Y.I.); kkito@ncc.go.jp (K.K.); moiwasak@ncc.go.jp (M.I.); stsugane@ncc.go.jp (S.T.); nsawada@ncc.go.jp (N.S.); 4Division of Preventive Medicine, Graduate School of Medical and Dental Sciences, Niigata University, 1-757 Asahimachidori, Niigata 951-8510, Japan; kazun@med.niigata-u.ac.jp; 5Department of Health Promotion Medicine, Graduate School of Medical and Dental Sciences, Niigata University, 1-757 Asahimachidori, Niigata 951-8510, Japan; juntatnk@med.niigata-u.ac.jp; 6Department of Public Health Medicine, Faculty of Medicine, Health Services Research and Development Center, University of Tsukuba, 1-1-1 Tennodai, Tsukuba 305-8575, Japan; yamagishi.kazumas.ge@u.tsukuba.ac.jp; 7Ibaraki Western Medical Center, 555 Otsuka, Chikusei 308-0813, Japan; 8Division of Epidemiology, National Cancer Center Institute for Cancer Control, 5-1-1 Tsukiji, Chuo-ku, Tokyo 104-0045, Japan; tyamaji@ncc.go.jp; 9Public Health, Department of Social Medicine, Graduate School of Medicine, Osaka University, 2-2 Yamadaoka, Suita-city, Osaka 565-0871, Japan; iso@pbhel.med.osaka-u.ac.jp; 10National Institutes of Biomedical Innovation, Health and Nutrition, 1-23-1 Toyama, Sinjuku, Tokyo 162-8636, Japan

**Keywords:** Food Frequency Questionnaire, screening, sodium, receiver-operating characteristic analysis, validity

## Abstract

Using Food Frequency Questionnaires (FFQs) to compare dietary references for screening has been in high demand. However, FFQs have been widely used for ranking individuals in a population based on their dietary intake. We determined the validity of sodium (salt equivalent) intake, potassium intake, and sodium-to-potassium (Na/K) ratio obtained using the FFQ for identifying individuals who deviated from the dietary reference intakes (DRIs) measured using multiple 24-h urinary excretion measurements or 12-day weighed food records (WFR). This study included 235 middle-aged subjects. The correlation coefficients (CCs) between the FFQ and WFR estimates were mostly moderate (0.24–0.54); the CCs between the FFQ and 24-h urinary excretion measurements were low or moderate (0.26–0.38). Values of area under the receiver-operating curve (AUC) at the point of DRIs for salt equivalent or potassium were >0.7 with the WFR as the reference standard and 0.60–0.76 with the 24-h urinary excretion as the reference standard. Using both standard measures, the AUC for the Na/K ratio was <0.7. The accuracy of salt equivalent and potassium intake estimation using the FFQ to determine absolute intake point was comparable to that using WFR, allowing for quantified error, but not as good as that of 24-h urinary excretion.

## 1. Introduction

Food Frequency Questionnaires (FFQs) have been widely used in large-scale epidemiological studies to evaluate the association between habitual diet and disease. FFQs rank individuals within the study population according to their intake status [1], and the validity of the ranking for this purpose has been reported [2,3,4,5]. However, the use of FFQs is in demand to assess the dietary habits of individuals compared with the dietary references for screening and nutrition education because FFQs exert fewer burdens on participants to estimate habitual intake. However, the feasibility of determining the absolute intake using FFQs and its accuracy has been scarcely examined in previous studies.

Diets that are high in sodium and low in potassium are associated with an increase in lifestyle-related diseases, such as hypertension [6,7,8], and a higher risk of cardiovascular disease [9]. The sodium and potassium intake levels of individuals worldwide are still higher than those recommended in the WHO guidelines, and reinforcement of the dietary management for the population is a must [10,11]. Moreover, the (urinary) sodium-to-potassium ratio (Na/K ratio) has recently been reported to be a useful index for assessing the dietary intake of reduced-sodium and increased-potassium [12,13]. The Na/K ratio has been positively associated with high blood pressure or hypertension in epidemiological studies [14], with its effect on systolic blood pressure being greater than sodium or potassium alone [6]. Thus, to achieve a reduced-sodium and increased-potassium diet in the population, it is important to develop screening tools that can help visualize the individual habitual dietary sodium and potassium intakes or the Na/K ratio compared with the dietary references.

An accurate measurement of an individual’s sodium and potassium intake is multiple 24-h urinary excretion collections; however, this method is inconvenient and requires labor [15]. Hence, weighed food records (WFR), which directly measure foods and amounts actually consumed, have been commonly used for national surveys in Japan. Therefore, in this study, we examined the validity of the absolute intakes of sodium (salt), potassium, and Na/K ratio as estimated using FFQs to detect intakes that deviate from the standard compared with the estimated intakes measured using 12-day weighed food records (12-d WFR) or multiple 24-h urinary excretion collection.

## 2. Materials and Methods

### 2.1. Study Design and Participants

The study was conducted in five areas designated by the Japan Public Health Center-based Prospective Study for the Next Generation (JPHC-NEXT) protocol (Yokote, Saku, Chikusei, Murakami, and Uonuma) [16]. The recruitment criteria were middle-aged and older residents of the five areas. A total of 255 generally healthy men and women voluntarily participated in this study [3]. Of the 253 participants who completed the survey, those who were not aged 40–74 years (*n* = 13) or who were unable to accomplish the collection of urinary excretion ≥3 times out of 5 times in a year (*n* = 5) were excluded. A total of 235 subjects (94 men and 141 women) were finally included in this study.

### 2.2. Ethics Approval and Consent to Participate

The study was approved by the Institutional Review Board of the National Cancer Center, Tokyo, Japan and all the other collaborating research institutions. The study was also approved by the Ethics committee of Nara Women’s University. This study was conducted in accordance with the Ethical standards of the 1964 Declaration of Helsinki and its later amendments. All the participants provided written informed consent before participation at the study settings.

### 2.3. Data Collection and Time Window

Reference intake data were obtained from all participants between November 2012 and December 2013 using 12-d WFR and 5 times’ collection of 24-h urinary excretion ([Fig nutrients-14-02594-g001]). To conduct the 12-d WFR, three-consecutive-day WFR data were obtained over four seasons at intervals of approximately 3 months. The 24-h urinary collection was self-administered on the last day of each three-consecutive-day WFR and at the end of the study. The self-administered semiquantitative FFQs (full version) were completed twice between November 2012 and December 2013 at a 1-year interval. Information on height, weight, and lifestyle, such as smoking and drinking habits, were collected using a self-report questionnaire combined with the second FFQ. To determine the validity of the estimated dietary intake based on this FFQ, information from the second administration was used because the FFQ contains questions regarding the individuals’ diet over the past year.

### 2.4. 12-d WFR

The 12-d WFR was conducted over a continuous duration of two weekdays and one weekend day at 3-month intervals across the four seasons. As a general rule, WFR was conducted on Thursday, Friday, and Saturday. Food portions were measured by each participant during the meal preparation using the supplied digital cooking scale (Tanita Co., Ltd., Tokyo, Japan) and measuring spoons and cups. For foods purchased or consumed outside the home, the participants were instructed to record the approximate quantity of all foods present in the meal and/or the names of the product and company. Trained dietitians assessed the dietary records with the participants on the day after each of the three-day WFR at the site in each study area and coded the records for foods and weights. For the intake assessed using the 12-d WFR, the mean of the 12 days was used.

### 2.5. 24-h Urinary Collection

The 24-h urinary collection was self-administered five times on the last day of each three-day-WFR and at the end of the study. Urinary specimens were collected using a portable urine measurement device (Urine Mate P, Sumitomo Bakelite Co., Ltd., Tokyo, Japan), which obtains a 1/50 portion of all collected urine. On the collection day, specimens obtained using the device were stored in a cold dark place and sent to a laboratory the next day. The urinary concentration of sodium and potassium (mEq/L) was analyzed by Kotobiken Medical Laboratories Inc. (Tokyo, Japan) using an ion-selective electrode method. Urinary collection with two or more errors (e.g., forgetting to conduct the sampling and spillage out of the container) at every 24 h was not included in the individual’s mean value [17]. A single urinary collection error was corrected by the mean value based on the individual’s collected urine volumes and the recorded number of collection times without an error. The subjects in whom an error in the collection occurred in more than three of the total five urinary collections within the study period were excluded from the analysis. The remaining subjects were eligible for analysis of 24-h urinary excretion of sodium and potassium. Urinary excretions for sodium and potassium were calculated using the following formula: 24-h urinary excretion (mg/day) = obtained (corrected) volume of urinary excretion (mL) × 50/1000 × urine concentration (mEq/L) × 23 for sodium, or ×39 for potassium. Subsequently, the individual’s mean urinary excretion values were calculated. Most subjects underwent 24-h urine collections 5 times, i.e., 86%, 11%, and 2% of the subjects underwent 24-h urine collections 5, 4, and 3 times, respectively. Furthermore, for converting urinary values to the corresponding dietary intakes (mg/day), they were multiplied by 1.0 and 1.3 for sodium and potassium intakes, respectively [18,19,20,21]. Furthermore, the Na/K obtained as a molar ratio calculated using intake values were multiplied by 1/1.3 for potassium intake so as to convert them to the corresponding urinary excretions.

### 2.6. FFQs

The FFQ (full version) includes 172 food and beverage items and nine frequency categories ranging from “almost never” to “seven or more times per day” (or “10 or more glasses per day” for beverages) and three portion size categories. The questionnaire comprises questions regarding the respondent’s usual consumption of the listed foods over the past year. The food list was initially developed for and used in the Japan Public Health Center-based prospective study; it was modified for middle-aged and older residents in several areas of Japan for use in the JPHC-NEXT Study baseline survey. The validity of the intake estimates based on the FFQ have been reported; the median of Spearman’s correlation coefficients (CCs) for energy and nutrients was 0.50 for men and 0.43 for women when compared with WFR [3]. Intakes were calculated using the Standard Tables of Food Composition in Japan 2010. These were also applied for the FFQ as a supplemental analysis.

### 2.7. Statistical Analysis

In this study, we examined sodium (salt equivalent, Na), potassium (K), and the Na/K ratio to assess the validity of the FFQ. The CCs between the intakes based on the FFQ and 24-h urinary excretion or 12-d WFR were analyzed for the purpose of validation to relatively rank the individuals. The CCs between the intakes based on 12-d WFR and 24-h urinary excretion were also calculated for comparison. The deattenuated energy-adjusted CCs were calculated using a residual model. Deattenuated CCs compared with the 12-d WFR were calculated based on the intraindividual variance from the usual intake, according to the 12-d WFR of each nutrient. Deattenuated CCs compared with 24-h urinary excretion were calculated based on the intraindividual variance, according to the five-time urinary collection among the subjects who completed the collection (*n* = 80 [85%] and *n* = 123 [87%] for men and women, respectively). Additionally, to examine the categorization agreement between the energy-adjusted estimation using the FFQ and that using the 24-h urinary excretion, we calculated the number of participants classified into the same, adjacent, and extreme categories by cross-classification according to quintile. The ability of the FFQ to estimate the absolute intake of Na (salt) and K and the Na/K ratio was assessed using a receiver-operating characteristic (ROC) analysis and by calculating the area under the ROC curve (AUC) and its 95% confidence intervals (95% CIs), with the definition based on a reference standard measured by the 12-d WFR or 24-h urinary excretion. As a reference standard, we used the values from Japanese Dietary Reference Intakes 2020 (DRIs) for each nutrient [22]. An AUC of >0.7 and a low limit of its 95% CI of >0.5 indicated that the FFQ-estimated intake could detect the deviation from the target value. The optimal cutoff values of the estimated intake by the FFQ were assessed as the optimal sensitivity and specificity achieved according to the maximal Youden’s index and minimum distance between the upper-left point and each point on the ROC curve. Sensitivity and specificity were also examined when the FFQ cutoff value was closest to the values of DRIs. The distance was calculated using the following formula: distance = (1 − sensitivity)^2^ + (1 − specificity)^2^. Moreover, we calculated the differences using the following formula: difference (%) = (cutoff value in the FFQ − criteria value based on each DRIs)/criteria value based on each DRIs × 100. Regarding the criteria values of DRIs, the value of “tentative dietary goal for preventing lifestyle-related diseases (DG),” which has been developed to prevent lifestyle-related diseases, was used for both Na and K, and the value of “adequate intake (AI),” which indicates the amount that is adequate to maintain a nutritional status, was used for K [22]. As the DRIs for the Na/K ratio have not been developed, its target value was defined as 2 in molar ratio in men and women based on previous research [13]. All analyses were performed using a SAS version 9.4 (SAS Institute Inc., Cary, NC, USA).

## 3. Results

### 3.1. Subject Characteristics

We included 94 men and 141 women in the main validity analysis. Table 1 shows the characteristics of the men and women included in the study. The mean (standard deviation) participant age was 57.3 (8.6) years for men and 57.1 (8.5) for women. The mean body mass index was 23.7 (2.8) kg/m^2^ for men and 22.8 (3.1) kg/m^2^ for women. The proportions of current smokers and heavy drinkers were 25.5% and 40.4% among men and 1.4% and 5.0% among women, respectively.

### 3.2. Correlation between Intake Estimated Using the FFQ and That Using 12-d WFR or 24-h Urinary Excretion

Table 2 shows the daily intake of Na and K, as well as the Na/K ratio as assessed by 24-h urinary excretion, 12-d WFR, and the FFQ and their correlations. The mean intakes of Na, K, and Na/K ratio estimated using the FFQ were similar to those using the 24-h urinary excretion or 12-d WFR in men; however, in women, the intakes of Na and K based on the FFQ were overestimated compared with those obtained using the 24-h urinary excretion or 12-d WFR. The deattenuated energy-adjusted CCs for the Na and K using the 12-d WFR with those using 24-h urinary excretion were moderate for men (r = 0.55 and 0.52, respectively) and relatively high (r = 0.71 and 0.67, respectively) for women. The CCs of Na/K for the 12-d WFR compared with the 24-h urinary excretion were the highest CCs among both the men and women (r = 0.76 and 0.87, respectively). The CCs of the estimated intakes using the FFQ with those using a 24-h urinary excretion were low or moderate (0.26–0.38), whereas the CCs of the estimated intakes using FFQ with those using the 12-d WFR were mostly moderate (0.24–0.54) in men and women. Regarding the agreement of classification using the FFQ and 24-h urinary excretion, the proportion of subjects classified into the opposite extreme categories was less than 5% for almost all intakes.

### 3.3. Validity of the Absolute Intake Estimated Using the FFQ to Determine the DRIs Compared with That of 12-d WFR or 24-h Urinary Excretion

Table 3 presents the results of the ROC curve for Na, K, and Na/K ratio using the FFQ compared with those using 24-h urinary excretion or 12-d WFR as the reference standard. Table 3 also displays the difference, sensitivity, and specificity at the closest cutoff value using the FFQ compared with the value of DRIs (in the upper part) in addition to those at the optimal cutoff value (in the lower part). When the 24-h urinary excretion is used as the reference standard, the AUC of the ROC curve for Na and K was slightly <0.7 for Na among women. Similarly, the AUC for K was slightly <0.7 in terms of AI and DG among men and DG among women. However, when the 12-d WFR was regarded as the standard measure, the AUC was >0.7, indicting moderate performance, irrespective of gender or type of reference. When the 12-d WFR was regarded as the standard reference, both the sensitivity and specificity at the optimal cutoff point were not <50%, but the degrees and directions of the differences from the reference value were not constant, depending on the gender and type of the references, as follows: for both nutrients, the differences were negative among men (−1.4% for Na, −1.9% for AI of K, and −13.6% for DG of K), whereas the differences were positive among women (6.2% for Na, 41.8% for AI of K, and 10.5% for DG of K). Based on the results using the 12-d WFR as a standard reference at the cutoff value by the FFQ, which is closest to the values of DRIs, the Youden’s index was less than that at an optimal cutoff point (ranging from 0.13 to 0.41 for the closest cutoffs and ranging from 0.35 to 0.46 for optimal cutoffs). When both the 24-h urinary excretion and the 12-d WFR were regarded as the standard reference (men and women combined), the AUC for the Na/K ratio was <0.6, and the lower limit of the 95% CI was <0.5. Additionally, Table 4 shows the results of the ROC curve using 12-d WFR compared with the 24-h urinary excretion as the standard reference. The AUC and Youden’s index at an optimal cutoff point for all nutrients were in a relatively higher range (0.78–0.97 and 0.47–0.87, respectively), and the differences from the value of DRIs was small (varied from 4.6% to 29.3%) compared with that of the FFQ exclusive of K in men.

## 4. Discussion

By calculating the AUC of the ROC curves and Youden’s index at the optimal cutoff values in addition to the deattenuated energy-adjusted CCs, we examined the validity of the absolute intakes of Na, K, and the Na/K ratio estimated using the FFQ to identify individuals who deviate from DRIs compared with the estimated intakes measured using the 12-d WFR or multiple 24-h urinary excretion. Although the CCs of the intake based on the FFQ with the 24-h urinary excretion were relatively low, those CCs with the 12-d WFR were moderate. The CCs between the intake based on the 12-d WFR and 24-h urinary excretion were relatively high. When 24-h urinary excretion was regarded as the standard reference, the AUC of the ROC curves was moderate depending on gender, DRIs, or nutrient. Conversely, when the 12-d WFR was regarded as the standard reference, all AUCs indicated moderate performance for both the Na and K, irrespective of sex. Even among those with moderate performance, the degrees and directions of the differences for the optimal cutoff points from the DRIs were not constant within the same nutrient.

In most of the validation studies conducted on FFQs, the validity of the relative values was examined using CCs between the nutrient intake, measured using the dietary record and that estimated using the FFQ. This was also examined using the agreement of the group divisions, such as the quartile or quintile divisions, for ranking the individuals [2,3]. The results of validation studies of Na estimation using the FFQ compared with those using the 24-h urinary excretion were also expressed as CCs [21]. Few studies have examined the validity of intakes by the FFQ or other questionnaires using ROC curves and/or AUC. Tsubono et al. [23] reported the accuracy of the FFQ as a screening test using the ROC curves compared with those of the three-day WFR using mean intakes based on the 9-day WFR as the standard reference. They reported moderately good performance for certain nutrients in the questionnaires, although that for Na was low when compared visually with the three-day WFR. In the present study, similar to previous study results obtained for certain nutrients, the FFQ showed moderately good performance for estimating Na and K intakes as analyzed by the ROC curve using the 12-d WFR as a standard reference.

In addition, four previous studies reported the performance of estimation using the FFQ [24] or a questionnaire specified to salt intake [25,26,27] based on the ROC curve and the AUC using a single 24-h urinary collection [24,26,27] or spot urinary collection [25] as the standard reference. Kelly et al. [24] validated the estimation of the FFQ compared with 24-h urinary sodium excretion using a cutoff point of ≥100 mmol/L for Na with an AUC of 0.76 (95% CI: 0.6, 0.9). Sasaki et al. [25] developed and validated a screening tool for salt intake using estimated 24-h urinary Na excretion from a single spot urine standard reference, with an AUC of 0.70 (95% CI: 0.67, 0.74) and a cutoff point of ≥12.2 g salt/day for men and ≥11.4 g salt/day for women. Monntoya et al. [26] evaluated the FFQ as a screening test for detecting excess salt intake (≥12 g/24 h) in French patients with hypertension and/or renal failure. They reported an AUC of 0.665 with 24-h urinary sodium excretion used as the standard reference. Mason et al. [27] developed a self-administered Na questionnaire for use in routine clinical care of Australian patients with chronic kidney disease and used the ROC curve analysis to assess the construct validity of the appropriated score cutoff points of the test. The AUC of the test was 0.713 (95% CI: 0.530, 0.896) using 24-h urinary Na excretion as the standard reference and a cutoff point of 100 mEq. In this study, regarding the comparison between 24-h urinary excretion and the FFQ for determining Na intake, the AUC in women was lower than that in previous reports, although the AUC among men was similar or higher. This might, in part, be due to an improved accuracy of the standard reference using the multiple 24-h urine samples in this study or a lower validity of the estimates of several nutrients or energy intake obtained from the FFQs among women, which might be explained by a difference in dietary variety between genders [3]. Additionally, to the best of our knowledge, no studies have examined the absolute intake of K or the Na/K ratio in addition to salt, which are assessed using the FFQ with urinary collections or the WFR as standard references. Moreover, our results showed the quantitative error at the cutoffs from the value of DRIs.

The validity of absolute intakes estimated using the FFQ was relatively low when compared with multiple 24-h urinary excretions, although higher CCs were observed than CCs reported in previous JPHC validation studies [28] and there were a few classifications in the extreme categories, even when FFQ was compared with 24-h urine excretion (which was similar to the comparison with 12-d WFR) [3]. The accuracy of the absolute intakes estimated using the FFQ for the purpose of determining the DRIs was not as good as that of 24-h urinary excretion. However, when 24-h urinary excretion was used as a standard reference, the AUC of the 12-d WFR was high, and the FFQ showed moderate performance when 12-d WFR was the standard reference. Thus, the FFQ can be considered for determining the DRIs, as it has a performance equivalent to that of 12-d WFR with the correction of a difference at an optimal cutoff point and with acceptance of the quantified error. Furthermore, the ROC of K was examined at two points of DRIs. The AUCs and degree of separation at cutoff values in the FFQ were different between these two points, thus indicating that it might not be possible to obtain a similar separation at any absolute point.

One of the limitations of this study is that the subjects were not selected by random sampling. Maintaining WFRs requires a high level of motivation, which might result in a larger population of health-conscious individuals than the general population [1], as partially indicated in the lower proportions of smokers in this study than that of the general Japanese population [29]. We cannot rule out the possibility that the results of this study overestimated the accuracy, although the total characteristics of the subjects indicate that they were not necessarily more motivated than the participants of the 2012 National Health and Nutrition Survey in Japan [29]. On the other hand, the proportion of heavy drinkers in men in this study was higher than that in the Japanese population [29]. The accuracy of the effect of drinking may be considerably reduced, owing to the underreporting of drinking, making the present results inaccurate. However, because the correlation coefficient of ethanol consumption between FFQ and WFR was high (r = 0.82) among the same population of our study [3], we do not consider that a high proportion of heavy drinkers affected the result. Next, this study included one subject who had a history of chronic renal failure and 39 (16.6%) subjects who were taking hypertension medication. However, creatinine levels [28] of the subject with a history of chronic renal failure was confirmed as appropriate in 4 out of 5 times of urine sampling, and the urine volumes [30] of the subjects who took antihypertensives with diuretics were also confirmed as appropriate. We did not exclude these subjects because we wanted to examine the screening performance of the FFQ among the general population, including those who take medication or have certain diseases. Moreover, the survey was conducted on specific days of the week. When considering dietary habits and their variety in another day of the week (i.e., foods captured by FFQ and not captured by WFR), the accuracy of FFQ might be attenuated in the present study. Furthermore, the ROC curve and AUC were based on certain arbitrary criteria from the Japanese DRIs, and the same result cannot be guaranteed when other criteria are used. Additionally, because both 12-d WFR and part of the salt intake assessment using the FFQ depends on food composition tables for calculating nutrient intakes (e.g., pickled vegetables or dried and salted fish), the correlation and estimated performance between the two, based on the ROC analysis, was possibly overestimated.

## 5. Conclusions

The estimation of salt equivalent and potassium using the FFQ to determine the point of absolute intake may be comparable with that of the WFR, allowing the quantified error, but it is not as accurate when compared with 24-h urinary excretion, except in the ranking of individuals. Considerable and careful attention is required when using FFQ estimation as an absolute intake for screening individuals.

## Figures and Tables

**Figure 1 nutrients-14-02594-g001:**
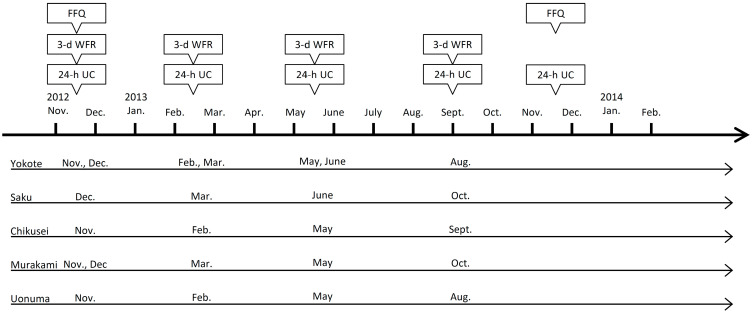
Data collection sequence in the validation study. d, days; FFQ, Food Frequency Questionnaire; WFR, weighted food record; UC, urinary collection.

**Table 1 nutrients-14-02594-t001:** Characteristics of the subjects (94 men and 141 women).

	Men		Women	
Age, years *	57.3	(8.6)	57.1	(8.5)
Body height, cm *	168.3	(6.9)	156.6	(5.7)
Body weight, kg *	67.3	(9.2)	56.0	(8.0)
BMI, kg/m² *	23.7	(2.8)	22.8	(3.1)
Current smoker, %	25.5		1.4	
Heavy drinker, % ^†^	40.4		5.0	

BMI, body mass index; * Values are reported as mean (standard deviation). ^†^ ≥280 and ≥140 g ethanol/week in men and women, respectively.

**Table 2 nutrients-14-02594-t002:** Comparison of sodium and potassium intake levels based on multiple 24-h urinary excretion measurements, FFQ, and 12-d WFR, as well as their CCs.

					CCs Compared with				Cross-Classification	
	Mean	(SD)	Median	Interquartile Range	24-h Urinary Excretion		12-d WFR		vs. 24-h Urine (%) ^||^	
									Same/	Extreme
					Crude	Adjusted *^,†^	Crude	Adjusted *^,†^	Adjacent	
Men (*n* = 94)										
Sodium, mg										
24-h urine	4650	(1196)	4589	3896–5350	-	-	-	-	-	-
12-d WFR	4579	(1112)	4453	3748–5283	0.61	0.55	-	-	70	0
FFQ	4378	(2061)	3918	2897–5529	0.33	0.36	0.37	0.33	67	3
Potassium, mg ^‡^										
24-h urine	2944	(874)	2832	2332–3433	-	-	-	-	-	-
12-d WFR	3113	(902)	3085	2527–3606	0.62	0.52	-	-	68	2
FFQ	3182	(1286)	2944	2253–3939	0.26	0.38	0.43	0.49	71	4
Na/K ratio ^§^, mmol/mmol										
24-h urine	3.68	(1.12)	3.42	2.86–4.40	-	-	-	-	-	-
12-d WFR	3.37	(0.80)	3.35	2.76–3.74	0.70	0.76	-	-	79	0
FFQ	3.04	(0.81)	2.96	2.50–3.56	0.30	0.32	0.22	0.24	65	5
Women (*n* = 141)										
Sodium, mg										
24-h urine	3922	(989)	3738	3336–4540	-	-	-	-	-	-
12-d WFR	3807	(924)	3697	3155–4314	0.64	0.71	-	-	75	0
FFQ	4508	(2060)	4031	3063–5833	0.17	0.26	0.30	0.41	62	7
Potassium, mg ^‡^										
24-h urine	3028	(900)	2956	2354–3607	-	-	-	-	-	-
12-d WFR	2979	(792)	2949	2389–3454	0.63	0.67	-	-	79	1
FFQ	3631	(1573)	3170	2479–4776	0.35	0.30	0.48	0.54	62	4
Na/K ratio ^§^, mmol/mmol										
24-h urine	3.03	(0.91)	2.99	2.33–3.55	-	-	-	-	-	-
12-d WFR	2.91	(0.71)	2.87	2.41–3.26	0.79	0.87	-	-	88	0
FFQ	2.78	(0.67)	2.74	2.32–3.07	0.27	0.30	0.39	0.42	62	4

12-d WFR, 12-day weighed food records; FFQ, Food Frequency Questionnaire; CC, correlation coefficient; SD, standard deviation; * Spearman’s rank correlation coefficients based on energy-adjusted values (other than Na/K ratio) and expressed as deattenuated CC. ^†^ Deattenuated CCx = observed CCx × SQRT (1 + λx/*n*), where λx is the ratio of within-to between-individual variance for number of dietary records. ^‡^ 24-h urinary potassium was adjusted to the intake level using the following formula: 24-h urinary potassium (mg) = 24-h urinary potassium excretion × 1.3; ^§^ The sodium/potassium (Na/K) ratio was adjusted to the urinary excretion level using 1/1.3 potassium for the 12-d WFR and FFQ intake levels. ^||^ The number of participants classified into the same, adjacent, and extreme categories using cross-classification according to quintile and energy-adjusted intake by the FFQ or 12-d WFR was compared to that using the 24-h urinary excretion.

**Table 3 nutrients-14-02594-t003:** Accuracy of the estimated salt and potassium intake levels using the FFQ to detect individuals with intake levels that deviated from the DRIs measured using multiple 24-h urinary excretion measurements or the 12-day WFR *.

	Criteria Based on DRIs	*n* *	AUC (95% CI)		Cutoff Values in the FFQ	%Dif ^†^	At Cutoff Value			
							Sensitivity, %	Specificity, %	Youden’s Index	Distance to Corner
Salt equivalent										
Multiple 24-h urinary excretion as reference standard										
men (*n* = 94)										
DG	≥7.5 g	87	0.76	(0.56–0.95)	7.48	−0.2	75	71	0.46	0.38
					9.32	24.3	63	86	0.49	0.39
Women (*n* = 141)										
DG	≥6.5 g	134	0.60	(0.37–0.82)	6.55	0.8	85	29	0.14	0.73
					9.93	52.8	54	71	0.26	0.54
12-d WFR as reference standard										
men (*n* = 94)										
DG	≥7.5 g	88	0.80	(0.62–0.98)	7.48	−0.2	74	67	0.41	0.42
					7.40	−1.4	76	67	0.43	0.41
Women (*n* = 141)										
DG	≥6.5 g	131	0.71	(0.50–0.91)	6.55	0.8	86	40	0.26	0.62
					6.90	6.2	86	60	0.46	0.42
Potassium ^‡^										
Multiple 24-h urinary excretion as reference standard										
Men (*n* = 94)										
AI	<2500 mg	34	0.60	(0.47–0.73)	2475	−1.0	50	73	0.23	0.57
					2591	3.6	56	72	0.28	0.52
DG	<3000 mg	57	0.66	(0.55–0.77)	3025	0.8	61	57	0.18	0.58
					2817	−6.1	60	70	0.30	0.50
Women (*n* = 141)										
AI	<2000 mg	14	0.71	(0.54–0.87)	1991	−0.5	36	90	0.25	0.65
					2625	31.2	71	75	0.46	0.38
DG	<2600 mg	47	0.68	(0.58–0.77)	2609	0.3	45	79	0.23	0.59
					2970	14.2	64	67	0.31	0.49
12-d WFR as reference standard										
Men (*n* = 94)										
AI	<2500 mg	23	0.73	(0.61–0.85)	2475	−1.0	61	73	0.34	0.47
					2452	−1.9	61	76	0.37	0.46
DG	<3000 mg	44	0.71	(0.60–0.81)	3025	0.8	66	56	0.22	0.56
					2591	−13.6	57	78	0.35	0.48
Women (*n* = 141)										
AI	<2000 mg	8	0.72	(0.57–0.86)	1991	−0.5	25	88	0.13	0.76
					2837	41.8	75	62	0.37	0.45
DG	<2600 mg	48	0.76	(0.68–0.85)	2609	0.3	52	83	0.35	0.51
					2874	10.5	71	74	0.45	0.39
Na/K ratio ^§^										
Men and women (*n* = 235)										
Multiple 24-h urinary excretion as reference standard										
--	≥2.00	215	0.59	(0.48–0.70)	2.01	0.5	91	10	0.01	0.90
	mmol/mmol				3.00	49.8	40	85	0.25	0.61
12-d WFR as reference standard										
--	≥2.00	224	0.57	(0.39–0.75)	2.01	0.5	91	18	0.09	0.82
	mmol/mmol				2.55	27.3	69	45	0.15	0.63

FFQ, Food Frequency Questionnaire; DRIs, Dietary Reference Intakes for Japanese, 2020; 12-d WFR, 12-day weighed food records; AUC, area under the curve; CI, confidence interval; DG, tentative dietary goal for preventing NCDs; AI, adequate intake; * The upper part of each term shows the results for the cutoff value in the FFQ that was the closest to the reference value, whereas the lower part shows the optimal cutoff value determined from Youden’s index and distance to corner. *n* *; Number of subjects who satisfied the criteria assessed using the reference standard. ^†^ Differences were calculated using the following formula: difference (%) = (cutoff value in the FFQ − criteria value based on each DRIs)/criteria value based on each DRIs × 100; ^‡^ 24-h urinary potassium was adjusted to the intake level using the following formula: 24-h urinary potassium (mg) = 24-h urinary potassium excretion × 1.3; ^§^ The sodium/potassium (Na/K) ratio was adjusted to the urinary excretion level using 1/1.3 potassium for the 12-d WFR and FFQ intake levels.

**Table 4 nutrients-14-02594-t004:** Accuracy of estimated salt and potassium intake levels using the 12-day WFR to detect individuals with intake levels that deviated from the DRIs measured using multiple 24-h urinary excretion measurements *.

	Criteria Based on DRIs	*n * ^†^	AUC (95% CI)		Cutoffs in the 12-d WFR	%Dif ^‡^	At Cutoff Value			
							Sensitivity, %	Specificity, %	Youden’s Index	Distance to Corner
Salt equivalent										
Men (*n* = 94)										
DG	≥7.5 g	87	0.90	(0.83–0.97)	7.51	0.1	95	29	0.24	0.72
					9.30	24.0	83	86	0.68	0.22
women (*n* = 141)										
DG	≥6.5 g	134	0.84	(0.75–0.93)	6.46	−0.7	94	14	0.08	0.86
					8.41	29.3	72	86	0.57	0.32
Potassium ^§^										
Men (*n* = 94)										
AI	<2500 mg	34	0.78	(0.67–0.88)	2527	1.1	53	90	0.43	0.48
					2879	15.2	74	77	0.50	0.35
DG	<3000 mg	57	0.82	(0.74–0.91)	3000	0.0	67	81	0.48	0.38
					3237	7.9	77	76	0.53	0.33
women (*n* = 141)										
AI	<2000 mg	14	0.93	(0.87–0.98)	2001	0.1	43	98	0.40	0.57
					2391	19.6	93	81	0.74	0.20
DG	<2600 mg	47	0.79	(0.72–0.87)	2605	0.2	60	78	0.37	0.46
					2719	4.6	72	74	0.47	0.38
Na/K ratio ^||^										
Men and women (*n* = 235)										
	≥2.00 mmol/mmol	215	0.97	(0.94–1.00)	2.00	0.0	100	50	0.50	0.50
					2.32	16.2	92	95	0.87	0.10

12-d WFR, 12-day weighed food records; DRIs, Dietary Reference Intakes for Japanese, 2020; AUC, Area under the curve; CI, confidence interval; DG, tentative dietary goal for preventing NCDs; AI, adequate intake. * The upper part of each term shows the results for the cutoff value in the 12-d WFR that was closest to the reference value, whereas the lower part shows the optimal cutoff value determined from Youden’s index and distance to corner. *n*
^†^; Number of subjects who satisfied the criteria assessed using the reference standard. ^‡^ Differences were calculated using the following formula: difference (%) = (cutoff value in the 12-d WFR − criteria value based on each DRIs)/criteria value based on each DRIs × 100. ^§^ 24-h urinary potassium was adjusted to the intake level using the following formula: 24-h urinary potassium (mg) = 24-h urinary potassium excretion × 1.3. ^||^ The sodium/potassium (Na/K) ratio was adjusted to the urinary excretion level using 1/1.3 potassium for the 12-d WFR intake level.

## Data Availability

According to ethical guidelines in Japan, we cannot publicly disclose individual data owing to participant privacy. Furthermore, the informed consent that we obtained does not include a provision for the data to be shared publicly. The datasets used and/or analyzed during the current study are available from the corresponding author on a reasonable request.

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
