# Peer review of "Validity of the Food Frequency Questionnaire—Estimated Intakes of Sodium, Potassium, and Sodium-to-Potassium Ratio for Screening at a Point of Absolute Intake among Middle-Aged and Older Japanese Adults"

_nutrients, 2022, doi:10.3390/nu14132594_

Round 1

Reviewer 1 Report

In the manuscript entitled: Validity of the Food Frequency Questionnaireestimated intakes of sodium, potassium, and sodium-to-potassium ratio for screening at a point of absolute intake compared with 12-day weighed food records or multiple 24-hour urinary excretion 5
among middle-aged and older Japanese adults, by T Matsuno et al., the authors determined the validity of sodium and potassium intake obtained using the FFQ for identifying individuals who deviated from the dietary reference intakes measured using multiple 24-h urinary excretion measurements or 12-day weighed food records (WFR). The major finding is that the accuracy of salt and potassium intake estimation using the FFQ, although comparable to that using WFR, was not as good as that of 24-h urinary excretion.

The study deals with an important issue of an accurate measurement of an individual’s sodium and potassium intake, in the face of the fact that the gold standard, i.e. the 24-h urinary excretion collection is inconvenient and troublesome to perform.

It is generally well designed and performed. The aim is clear, methodological tools appropriate and the results convincing. The manuscript is well written and easy to follow. The limitations of the study are recognized and presented.

The authors may wish to consider the following issues in the revision of their paper.

1.      For sodium and potassium intake, it is the 24-hour urine collection that is acknowledged as the gold standard ( J Hum Hypertens 33, 345–348; 2019). Just for mu knowledge, but also for the readers, why did the authors include also the WHR in their analysis?

2.      Do the authors possess any data on kidney failure and/or use of diuretics, states that might alter the results of the 24-hour urine collection?

Author Response

Thank you for your kind critique of our manuscript. We have revised the manuscript according to your comments. Please see the attachment.

Reviewer 2 Report

In the present review, Tomoka Matsunois et al examined the validity of the Food Frequency Questionnaire-estimated in- 2 takes of sodium, potassium, and sodium-to-potassium ratio for 3 screening at a point of absolute intake compared with 12-day 4 weighed food records or multiple 24-hour urinary excretion 5 among middle-aged and older Japanese adults. The paper is very interesting and correctly written. However, the data on which is out of date. Also, the main publications to which the authors refer seem to be very old.  I addressed a few issues below:

1.   The title seems a bit too long and could therefore be confusing.

2.      In the introduction the latest reference is of 2017. The authors should add references to the latest studies.

3.      Details should be added in the Methods section. References to previous studies seem to reduce transparency.

4.      Please explain the reason for the disparity between men and women.

5.      The method itself was difficult to perform and subject to a high risk of error.

6.      In the group of men, 40% drank a lot of alcohol and 25% smoked cigarettes, which may have a major impact on the outcome. Are these proportions typical of the population under study?

7.      It is very difficult to determine the composition of meals outside the home. whether the authors have indicated the number/proportion of meals outside the home?

8.      The authors write: “The 12-d WFR was conducted over a continuous duration of two weekdays and one weekend day at 3-month intervals across the four seasons”. Was it always the same days? People often have dietary habits during the week - such as a hearty lunch on Sunday that they eat on Monday, so they are likely to differ, given Thursday and Monday, for example. I have not found any information on whether these days change or whether they are random.
